# Grouped Sequential Optimization Strategy - the Application of Hyperparameter Importance Assessment in Deep Learning

Ruinan Wang, Ian Nabney, and Mohammad Golbabaee

Department of Engineering Mathematics, University of Bristol, United Kingdom

{zg21696, in17746, an22148}@bristol.ac.uk

Hyperparameter optimization (HPO) is a critical component of machine learning pipelines, significantly affecting model robustness, stability, and generalization. However, HPO is often a time-consuming and computationally intensive task. Traditional HPO methods, such as grid search and random search, often suffer from inefficiency. Bayesian optimization, while more efficient, still struggles with high-dimensional search spaces. In this paper, we contribute to the field by exploring how insights gained from hyperparameter importance assessment (HIA) can be leveraged to accelerate HPO, reducing both time and computational resources. Building on prior work that quantified hyperparameter importance by evaluating 10 hyperparameters on CNNs using 10 common image classification datasets, we implement a novel HPO strategy called 'Sequential Grouping.' That prior work assessed the importance weights of the investigated hyperparameters based on their influence on model performance, providing valuable insights that we leverage to optimize our HPO process. Our experiments, validated across six additional image classification datasets, demonstrate that incorporating hyperparameter importance assessment (HIA) can significantly accelerate HPO without compromising model performance, reducing optimization time by an average of 31.9% compared to the conventional simultaneous strategy.

## 1. Introduction

In recent years, the rapid advancement of deep learning has led to significant breakthroughs across a wide range of applications, from computer vision to natural language processing, where hyperparameter optimization (HPO) has become increasingly vital in constructing models that achieve optimal performance. As the demand for HPO has been growing, the computational and time costs associated with it have become a significant bottleneck [1]. In this context, Hyperparameter Importance Assessment (HIA) has emerged as a promising solution. By evaluating the importance weights of individual hyperparameters and their combinations within specific models, HIA provides valuable insights into which hyperparameters most significantly impact model performance [2]. With this understanding, deep learning practitioners can focus on optimizing only those hyperparameters that have a more pronounced effect on performance. For less critical hyperparameters, users can reduce the search space during optimization or even fix them at certain values, thereby saving time in the model optimization process [3]. Although there has been considerable exploration of HIA, most existing studies have primarily focused on introducing new HIA methods or determining the importance rankings of hyperparameters for specific models within certain application scenarios. However, there has been limited exploration of how these insights can be strategically applied to enhance the efficiency of the optimization process.

To address the challenges in the current research landscape, this paper aims to use Convolutional Neural Networks (CNNs) as the research case to introduce HIA into the deep learning pipeline, demonstrating that the insights gained from HIA can effectively enhance the efficiency of hyper-

Second Conference on Parsimony and Learning (CPAL 2025).

parameter optimization. On a deeper level, this approach can also contribute to a more profound understanding and increased transparency of these "black box" models to some extent. From a practical perspective, it has the potential to become a foundational component for constructing machine learning pipelines. This could foster advancements in automated machine learning and contribute to the development of more interpretable and efficient deep learning models.

In this paper, the main contributions are as follows:

- We propose a novel Grouped Sequential optimization strategy (GSOS), which leverages hyperparameter importance weights obtained through Hyperparameter Importance Assessment (HIA) on CNN models. These importance weights, derived from the evaluation of 10,000 models trained across 10 datasets, provide constructive guidance for hyperparameter grouping and optimization sequencing.
- We integrate this strategy into Tree-structured Parzen Estimator (TPE)-based Bayesian optimization, which is a well-established hyperparameter optimization method introduced by Bergstra et al. [4]. Building on this foundation, we validate our GSOS strategy across six image classification datasets. To ensure robust results and mitigate the effect of randomness from initial sampling, we conduct five independent runs of HPO on each dataset, with 100 iterations per run. Our results show that the Grouped Sequential strategy reduces the time to find optimal hyperparameters by 19.69% and the overall optimization time by 31.90% compared to traditional simultaneous strategies.

## 2. Related Work

Many Hyperparameter Importance Assessment (HIA) studies draw inspiration from Feature Selection techniques, which aim to reduce computational costs by identifying impactful input features [5]. While Feature Selection focuses on input features, HIA targets hyperparameters. Early HIA methods, such as Forward Selection and Functional ANOVA, have high time complexity [6]. This prompted the development of N-RReliefF to reduce computational time while maintaining ranking consistency [7]. N-RReliefF has shown efficiency advantages without sacrificing quality, with studies indicating consistent importance rankings for hyperparameters such as "gamma" for SVM and "split criterion" for Random Forest [2, 7].

In deep learning, HIA research has followed two main directions. The first direction involves introducing new HIA methods, ensuring they match or improve upon established baselines, such as Functional ANOVA, in terms of both consistency and efficiency. For instance, the Plackett-Burman-based QIM method achieved similar hyperparameter rankings to Functional ANOVA on CIFAR-10 and MNIST datasets but improved computational efficiency threefold [8]. Other work has leveraged sensitivity analysis frameworks based on Morris and Sobol methods, showing that hyperparameters such as learning rate decay and batch size play crucial roles in complex datasets like CIFAR-10 [9]. Recently, HyperDeco-based e-AutoGR effectively identified hyperparameter importance for graph representation learning by removing confounding effects [10]. N-RReliefF has also been applied to deep learning, confirming the influence of hyperparameters such as number of convolutional layers, learning rate, and dropout rate on model performance [11].

The second direction applies HIA to specific deep learning models, such as ResNet and quantum neural networks (QNNs). Functional ANOVA identified the most impactful hyperparameters for ResNet across image classification datasets, including learning rate and weight decay [12]. For QNNs, key hyperparameters such as learning rate and data encoding strategy have been shown to influence performance on small datasets [13]. Additionally, a Bayesian optimization with Hyperband (BOHB) approach was employed to jointly search for neural architecture and hyperparameters, demonstrating performance gains on CIFAR-10 and reduced computational costs [14].

While much of HIA research focuses on identifying hyperparameter rankings, there is limited work on directly leveraging these insights to improve hyperparameter optimization efficiency. Our study addresses this gap by integrating HIA results into the optimization workflow, demonstrating that

a strategic application of HIA insights can lead to efficient model tuning without compromising performance.

## 3. TPE-based Bayesian Optimization

This section does not present TPE-based Bayesian optimization as a novel concept; instead, it builds upon the foundational work of Bergstra et al. and provides a more detailed derivation and operational description in this chapter to enhance understanding and facilitate practical application of the algorithm [4].

### 3.1. Acquisition Function in TPE Algorithm

TPE-based BO models the objective function by dividing the search space into two regions, represented by the "good distribution" $l(x)$ and the "bad distribution" $g(x)$, estimated using Parzen window density estimation based on observed objective values [15]. The data are partitioned into favourable and unfavourable regions based on a quantile threshold $\gamma$.

Qualitatively, the "good region" represents hyperparameter configurations that have shown promising performance, i.e., achieving objective values below the threshold $y^*$. This region corresponds to areas in the search space that are likely to yield better model performance and are therefore prioritized during optimization. Conversely, the "bad region" includes configurations with objective values above $y^*$, which are less likely to improve the objective and serve primarily as a reference for guiding the search away from suboptimal areas. By focusing exploration on the "good region" while maintaining awareness of the "bad region," the TPE algorithm effectively balances the exploitation of promising configurations with the broader exploration of the search space.

The acquisition function in TPE-based Bayesian optimization is reformulated from Expected Improvement (EI) to maximize the ratio between the good and bad distributions, guiding the search toward more promising regions. Starting from the EI formula, EI can be rewritten using Bayes' theorem as $\mathrm{EI}_{y^*}(x) = \int_{-\infty}^{y^*}(y^*-y)p(y|x)\,dy = \int_{-\infty}^{y^*}(y^*-y)\frac{p(x|y)p(y)}{p(x)}\,dy$.

Here, $\mathrm{EI}_{y^*}(x)$ represents the expected improvement for $y$ over the threshold $y^*$ for a configuration $x$, where $y < y^*$ represents an improvement extent. Typically, $y^*$ is set by a threshold proportion $\gamma$ (often 0.15 or 0.25), and the new configuration $x^*$ is then selected by maximizing EI: $x^* = \arg\max_x \mathrm{EI}_{y^*}(x)$.

To partition the search space based on $\gamma$, we redefine $p(x|y)$ as $l(x) = p(x|y)$ if $y < y^*$ and $g(x) = p(x|y)$ if $y > y^*$.

Next, we relate $l(x)$, $g(x)$, and $p(x|y)$ through $p(x)$:

$$p(x) = \int_R p(x|y)p(y)\,dy \tag{1}$$

$$= \int_{-\infty}^{y^*} l(x)p(y)\,dy + \int_{y^*}^{+\infty} g(x)p(y)\,dy \tag{2}$$

$$= \gamma l(x) + (1-\gamma)g(x) \tag{3}$$

Substituting this expression into the equation for $\mathrm{EI}_{y^*}(x)$, we can obtain:

$$\mathrm{EI}_{y^*}(x) = \int_{-\infty}^{y^*} (y^* - y) \frac{p(x|y)p(y)}{p(x)}\, dy \tag{4}$$

$$= \int_{-\infty}^{y^*} (y^* - y) \frac{l(x)p(y)}{\gamma l(x) + (1-\gamma)g(x)}\, dy \tag{5}$$

$$= \frac{l(x)}{\gamma l(x) + (1-\gamma)g(x)} \int_{-\infty}^{y^*} (y^* - y)p(y)\, dy \tag{6}$$

To maximize $\mathrm{EI}_{y^*}(x)$, we focus on the term containing $x$, yielding $\mathrm{EI}_{y^*}(x) \propto \left(\gamma + \frac{g(x)}{l(x)}(1-\gamma)\right)^{-1}$. Since $\gamma$ is constant, this result shows that the acquisition function in the TPE algorithm favours candidate points closer to the good distribution $l(x)$ and further from $g(x)$.

### 3.2. Optimization Algorithm Steps and Parzen Window Density Estimation

As discussed previously, Parzen Window Density Estimation is used to construct the "good" distribution $l(x)$ and the "bad" distribution $g(x)$. In this work, we apply a smooth kernel function, specifically the Gaussian kernel, to estimate the density, which is commonly referred to as Kernel Density Estimation (KDE) in the context of this algorithm.

The KDE is computed as follows:

$$\mathrm{KDE}(x) = \frac{1}{nh} \sum_{i=1}^{n} K\left(\frac{x - x_i}{h}\right)$$

where $n$ is the total number of samples, $h$ is the bandwidth parameter controlling the smoothness of the density estimate (a larger $h$ results in a smoother estimate), $x_i$ is the $i$-th observation in the sample data, and $K\left(\frac{x-x_i}{h}\right)$ is the kernel function that determines the influence of each sample point $x_i$ on the estimate at $x$. Here, we use the Gaussian kernel function, defined as

$$K(u) = \frac{1}{\sqrt{2\pi}} \exp\left(-\frac{u^2}{2}\right)$$

which provides a smooth, bell-shaped influence for each sample.

This approach provides an efficient mechanism for exploration and exploitation in high-dimensional, often non-convex hyperparameter spaces. The algorithm 1 outlines these steps in detail.

In the algorithm 1, since each hyperparameter $H_j$ is assumed to be independent of each other, we can model each $H_j$'s KDE based on $S_{\mathrm{good}}$ and $S_{\mathrm{bad}}$, and stack them together to form the joint distribution over the entire hyperparameter configuration, enabling us to find the optimal hyperparameters that improve the objective function.

$$x^* = \arg\max_x \prod_{j=1}^{J} \frac{l(h_j)}{g(h_j)} = \arg\max_x \frac{l(x)}{g(x)} \tag{7}$$

## 4. Grouped Sequential Optimization Strategy

GSOS leverages hyperparameter importance values to sequentially optimize groups of hyperparameters in a structured manner. This approach groups hyperparameters by importance, with more impactful parameters being optimized earlier in the sequence. After optimizing each group, the best values from that group are incorporated into the default hyperparameter configuration, allowing subsequent groups to benefit from previously optimized values. This strategy effectively balances efficiency and performance by focusing on the most critical hyperparameters first while retaining flexibility for subsequent optimizations.

---

**Algorithm 1** TPE Optimization

---

1: **Input:** Objective function $f(x)$, search space of hyperparameters $P = \{H_1, H_2, \ldots, H_J\}$, max iterations $m$, initial sample size $n = 15$, quantile $r = 0.25$
2: **Output:** Optimized hyperparameter configuration $x_{opt} = \{h_{opt1}, h_{opt2}, ..., h_{opt_J}\}$ that minimizes $f(x)$
3: Initialize sample set $S = \emptyset$
4: **for** $i = 1$ to $n$ **do**
5:     Randomly sample $x_i$ from $P$ and evaluate $f(x_i)$
6:     Add $(x_i, f(x_i))$ to $S$
7: **end for**
8: **for** iter $= 1$ to $m$ **do**
9:     Sort $S$ by $f(x)$ in ascending order
10:     Get threshold $y^*$ based on the $r$-th quantile of $f(x)$ values in $S$
11:     $S_{\text{good}} = \{x \mid f(x) \leq y^*\}$
12:     $S_{\text{bad}} = \{x \mid f(x) > y^*\}$
13:     Initialize $l(x) = 1$ and $g(x) = 1$
14:     **for** each hyperparameter $H_j$ in $P$ **do**
15:         Construct $l(h_j)$ using KDE on $H_j$ values in $S_{\text{good}}$
16:         Construct $g(h_j)$ using KDE on $H_j$ values in $S_{\text{bad}}$
17:         $l(x) = l(x) \cdot l(h_j)$
18:         $g(x) = g(x) \cdot g(h_j)$
19:     **end for**
20:     Select $x^* = \arg\max_x \left(\gamma + \frac{g(x)}{l(x)}(1 - \gamma)\right)^{-1}$
21:     Evaluate $f(x^*)$
22:     Add $(x^*, f(x^*))$ to $S$
23: **end for**

---

 

---

**Algorithm 2** Grouped Sequential Optimization Strategy (GSOS)

---

1: **Input:** Objective function $f(x)$, search spaces of each hyperparameter group $\{P_1, P_2, \ldots, P_K\}$ sorted by importance, max iterations for each group $\{m_1, m_2, \ldots, m_K\}$, default hyperparameter configuration $x_{\text{default}} = \{x_1, x_2, \ldots, x_D\}$
2: **Output:** Optimized hyperparameter configuration $x_{\text{optimal}} = \{x_{opt1}, x_{opt2}, \ldots, x_{opt_D}\}$
3: Initialize $x_{\text{current}} = x_{\text{default}}$
4: **for** each group $P_k$ in $\{P_1, P_2, \ldots, P_K\}$ **do**
5:     Define $f_{P_k}(x) = f(x)$ with hyperparameters outside $P_k$ fixed to $x_{\text{current}}$
6:     Optimize $P_k$ using TPE to obtain $x^*_{P_k} = $ TPE Optimization$(f_{P_k}(x), m_k, P_k)$
7:     Update $x_{\text{current}}$ with $x^*_{P_k}$
8: **end for**
9: Set $x_{\text{optimal}} = x_{\text{current}}$

---

To facilitate understanding, we have provided explanations for some key steps in Algorithm 2.

**Line 5 (Objective Function Set Up):** In each iteration, the objective function $f_{P_k}(x)$ needs to be reset up, where only the hyperparameters in the current group $P_k$ are the optimized target. All other hyperparameters are fixed with the corresponding values in $x_{\text{current}}$.

**Line 6 (TPE Optimization):** Alogrithm1 is applied to $f_{P_k}(x)$, with $P_k$ as the search space and $m_k$ as the maximum number of iterations. This produces the optimal values $x^*_{P_k}$ for the hyperparameters within the group $P_k$.

**Line 7 (Updating the Current Configuration):** The optimized values $x^*_{P_k}$ for the group $P_k$ are then used to update the corresponding hyperparameters in $x_{\text{current}}$. This ensures that subsequent groups are optimized based on the current best configuration, iteratively improving the overall hyperparameter setup.

This demonstrates how GSOS serves as an outer framework, sequentially invoking the TPE-based Bayesian Optimization subroutine to optimize each hyperparameter group. After completing the optimization of each group, the optimal hyperparameter values for that group are updated in the global configuration, allowing subsequent groups to be optimized based on the current best configuration.

# 5. Experimental Setup

Note that all experiments are conducted on a system equipped with five NVIDIA GeForce RTX 3090 GPUs and 64 GB of RAM, providing robust computational resources for efficient training and optimization.

## 5.1. Grouping Hyperparameters

To establish a solid foundation for hyperparameter grouping and sequential optimization, we reference the hyperparameter importance weights from the study by Wang et al. [11]. This study evaluated the relative impact of various hyperparameters on CNN performance, providing quantitative importance scores based on empirical experiments.

Figure 1 illustrates the normalized importance weights of the investigated CNN hyperparameters, organized into three distinct groups based on their relative impact. The horizontal axis represents the normalized weights on a [0,1] scale, providing a clearer view of the differences in hyperparameter influence across groups. The number of convolutional layers emerges as the most influential hyperparameter (weight: 0.385), underscoring the critical role it plays in CNN optimization. The learning rate

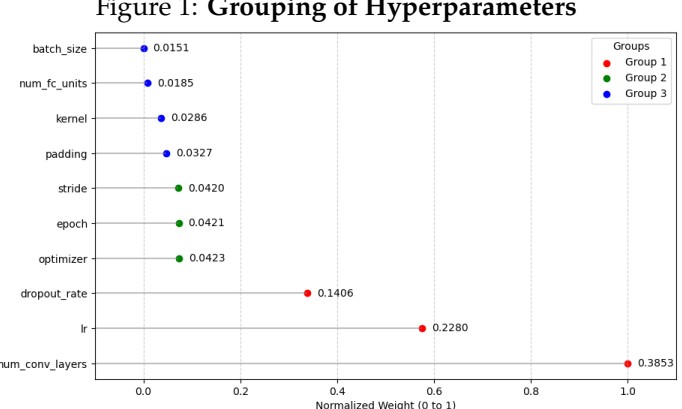

Figure 1: **Grouping of Hyperparameters**

closely follows a weight of 0.228, aligning with well-established research that emphasizes the importance of a carefully tuned learning rate for model convergence and stability. The dropout rate, with the third highest importance weight of 0.131, further demonstrates the necessity of regularization in preventing overfitting, especially for complex models. Other hyperparameters, including the optimizer type and the number of epochs, have comparatively lower weights but still contribute to the training dynamics. Conversely, parameters with the lowest weights, such as batch size (0.015), exhibit minimal impact on performance, suggesting that these parameters can be optimized later to prioritize computational resources for higher-impact hyperparameters. Moreover, these results affirm common practices in machine learning and align well with empirical insights, validating the heuristic strategies often used by practitioners during hyperparameter tuning.

## 5.2. Search Space and Model Architecture Limitation

The hyperparameter search space is defined to explore various configurations within a specified range. Table 1 outlines the search space for each hyperparameter, including the range or set of values being searched, and its default value. In this search space, each hyperparameter's value range was chosen to balance exploration efficiency and computational feasibility. The default values represent a

Table 1: **Hyperparameter Search Space**

| Variable | Search Space | Default Value |
|---|---|---|
| num_conv_layers | [2, 3, 4] | 3 |
| lr | [1e-5, 1] | 0.01 |
| dropout_rate | [0, 0.9] | 0.0 |
| optimizer | {adam, sgd} | adam |
| epoch | [10 to 100] | 10 |
| stride | [1, 2] | 1 |
| padding | {valid, same} | same |
| kernel | [3, 5] | 3 |
| num_fc_units | [64 to 256] | 64 |
| batch_size | [32, 64, 128, 256] | 32 |

baseline configuration, providing a starting point for
the optimization process.

The architecture of the CNN model adapts dynamically throughout while preserving several core design principles. Specifically: (1) Each convolutional layer is immediately followed by a ReLU activation and a pooling layer. If the structural hyperparameter, specifically the number of convolutional layers, is set to more than one, each additional convolutional layer is followed by a corresponding ReLU and pooling layer. (2) All pooling layers consistently use a max-pooling approach. (3) The final output layer employs a Softmax function to generate predictions. (4) Prior to the Softmax layer, the network includes two identical blocks in sequence, each consisting of a dropout layer followed by a fully connected layer, with the same dropout rate applied across both dropout layers. (5) A ReLU activation follows the first fully connected layer. (6) In general, unless stated otherwise, hyperparameters such as padding, stride, and dropout rate are uniformly configured across all relevant layers.

## 5.3. Datasets

In this study, hyperparameter optimization was performed using both GSOS and a baseline method on six different datasets: AHE [16], Intel Natural Scene [17], Rock Paper Scissors [18], Dog Cat [18], Flowers [19], and Kuzushiji-49 [20]. These datasets span various domains and offer diversity in terms of data volume, image dimensions, class count, and colour channels, making them well-suited for evaluating the general applicability of GSOS. If GSOS can demonstrate efficiency across all these datasets compared to traditional parallel hyperparameter optimization techniques, it would support its broader applicability across different data types. Table 2 provides an overview of the key characteristics of each dataset.

Table 2: **Summary of Datasets Used for Hyperparameter Optimization**

| Dataset | Number of Data Points | Image Dimensions | Number of Classes | Color Channels |
|---|---|---|---|---|
| AHE [16] | 7,599 | 224x224 | 4 | 3 |
| Intel Natural Scene [17] | 25,000 | 150x150 | 6 | 3 |
| Rock Paper Scissors [18] | 2,892 | 300x300 | 3 | 3 |
| Dog Cat [21] | 25,000 | 256x256 | 2 | 3 |
| Flowers [19] | 4,242 | 224x224 | 5 | 3 |
| Kuzushiji-49 [20] | 270,912 | 28x28 | 49 | 1 |

## 5.4. Experimental Group vs. Control Group

To evaluate the effectiveness of the GSOS compared to traditional parallel Bayesian optimization, we designed an experimental group and a control group. In Bayesian optimization, the initial sampling within the search space is inherently stochastic, which introduces a degree of variability in the results due to randomness. To mitigate this, both the experimental and control groups will execute multiple rounds on each dataset to provide a robust comparison.

For the experimental group, we employ the GSOS approach on the six datasets, with each trial running 100 iterations across five rounds. Within each round, the hyperparameters are grouped by importance into Group 1, Group 2, and Group 3, with iteration ratios of 4:3:3, respectively. This prioritization of iteration counts allows the strategy to focus more on high-impact hyperparameters early in the optimization process. For each iteration in every round, we record accuracy, loss, and time spent. These metrics will be further analyzed to assess the efficiency and stability of GSOS across diverse datasets. In contrast, the control group applies traditional parallel Bayesian optimization on the same six datasets, with the same configuration of 100 iterations across five rounds. This approach treats all hyperparameters equally in parallel without grouping, thus serving as a baseline to evaluate the added value of GSOS's sequential grouping.

By conducting these comparative experiments on identical datasets with consistent iteration settings, we aim to observe and analyze whether GSOS achieves higher efficiency and better performance stability than the conventional parallel approach.

# 6. Evaluation and Results

## 6.1. Comparison of Optimization Strategies

Table 3: **Comparison Results on Various Datasets**

| Dataset | Strategy | Avg Time to Find the Optimal HP | Avg Optimizing Time | Avg Val Accuracy | Avg Test Accuracy |
|---|---|---|---|---|---|
| AHE | grouped_sequential | 00:27:19 | 00:37:22 | 0.7433 | 0.7267 |
| AHE | simultaneous | 00:37:37 | 00:52:00 | 0.7646 | 0.7298 |
| Intel_Natural_Scene | grouped_sequential | 00:22:28 | 00:51:05 | 0.7369 | 0.7605 |
| Intel_Natural_Scene | simultaneous | 00:49:21 | 01:18:32 | 0.7647 | 0.7513 |
| Rock_Paper_Scissors | grouped_sequential | 00:22:10 | 00:32:18 | 0.9890 | 0.9784 |
| Rock_Paper_Scissors | simultaneous | 00:19:11 | 00:51:38 | 0.9959 | 0.9858 |
| Dog_Cat | grouped_sequential | 01:32:01 | 02:13:30 | 0.7577 | 0.7404 |
| Dog_Cat | simultaneous | 02:01:16 | 03:01:09 | 0.7929 | 0.7538 |
| Flowers | grouped_sequential | 00:48:57 | 01:04:20 | 0.6665 | 0.6651 |
| Flowers | simultaneous | 01:10:31 | 01:32:15 | 0.6823 | 0.6669 |
| Kuzushiji_49 | grouped_sequential | 03:04:42 | 03:26:41 | 0.9613 | 0.9260 |
| Kuzushiji_49 | simultaneous | 03:17:11 | 05:15:47 | 0.9650 | 0.9307 |

In this section, we analyze the effectiveness of the Grouped Sequential Optimization Strategy in comparison to the traditional simultaneous hyperparameter optimization approach. Our comparison focuses on four core metrics: Average Time to Find the Optimal Hyperparameters, Total Optimization Time, Validation Accuracy, and Test Accuracy. Table 3 presents these metrics, averaged over five runs for each dataset and strategy. The results indicate that GSOS consistently outperforms the simultaneous approach in terms of time efficiency across all datasets, and achieves faster convergence toward optimal configurations. However, there is a slight trade-off in accuracy, with minor reductions in validation and test accuracy observed for GSOS in most datasets.

## 6.2. Analysis of Time Reduction and Accuracy Change

To quantify the advantages of GSOS in terms of time efficiency, Table 4 summarizes the percentage reduction in optimization time and the changes in validation and test accuracy for GSOS compared to the simul-

Table 4: **Time Reduction and Accuracy Change**

| Dataset | Time Reduction (%) | Val Accuracy Change | Test Accuracy Change |
|---|---|---|---|
| AHE | 28.141 | -0.02133 | -0.00308 |
| Intel_Natural_Scene | 34.953 | -0.02781 | 0.00019 |
| Rock_Paper_Scissors | 37.444 | -0.00685 | -0.0074 |
| Dog_Cat | 26.304 | -0.0352 | -0.01346 |
| Flowers | 30.262 | -0.01576 | -0.00183 |
| Kuzushiji_49 | 34.549 | -0.00374 | -0.00476 |

taneous approach. On average, GSOS achieves a 19.69% faster time to find the optimal hyperparameters and a 31.90% reduction in the total optimization time. However, there is a trade-off with accuracy, as GSOS results in an average decrease of 2.23% in validation accuracy and 0.44% in test accuracy compared to the simultaneous approach. This trade-off highlights the potential of GSOS for applications where optimization time is a priority and minor reductions in accuracy are acceptable. The results demonstrate that GSOS offers a significant time-saving advantage while maintaining competitive performance levels across various datasets, making it a viable alternative for hyperparameter optimization in time-sensitive scenarios.

## 6.3. Analysis of Search Efficiency Across Datasets between GSOS and Simultaneous Strategy

Figure 2 presents a comparative analysis of GSOS and Simultaneous Strategy across multiple datasets, showing the progression of accuracy over iterations. The colour gradient in Figure 2 visually highlights the concentration of evaluations over iterations, with red representing high-density regions and blue indicating low-density regions. In GSOS, high-density areas (red regions) often have two: early iterations tend to focus on low-performance regions, while later iterations concentrate on high-performance regions. In contrast, the Simultaneous strategy usually displays a single, less intense high-density region concentrated in the high-performance area. The red colour in this region is lighter compared to GSOS's red zones, suggesting that the Simultaneous approach, although achieving concentration, explores with a lower density and less refinement. This pattern indicates broader, less focused exploration throughout the search space, which may sometimes limit its efficiency in finding optimal hyperparameter configurations compared to the GSOS approach.

Figure 2: **Scatter plot comparisons of the two strategies (Accuracy vs. Iterations).**

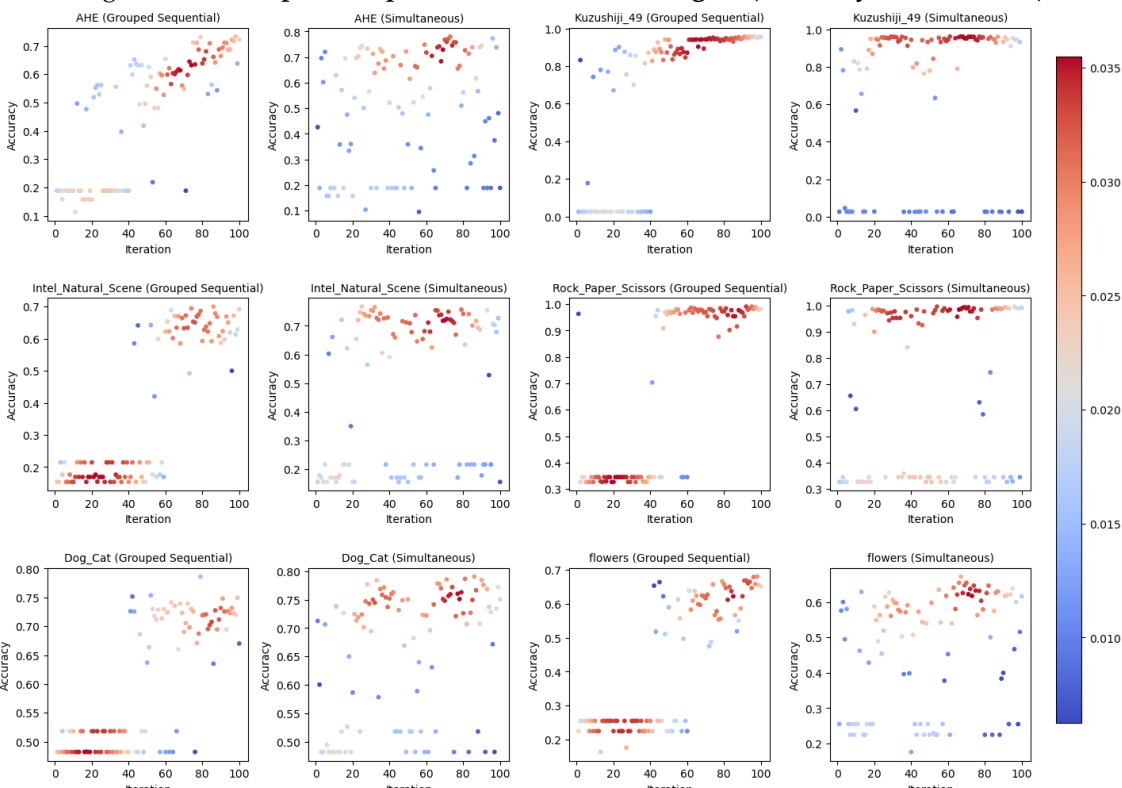

Across most datasets, GSOS achieves higher accuracy levels more quickly. In datasets like *AHE* and *Intel Natural Scene*, GSOS shows significant accuracy improvements within the first 50 iterations, while the Simultaneous approach requires more iterations to reach similar accuracy. Furthermore, in datasets such as *Dog Cat* and *Flowers*, GSOS displays a more stable and consistent improvement in accuracy, contrasting with the Simultaneous strategy, which exhibits greater fluctuation and scattered accuracy values across iterations.

Overall, these results indicate that GSOS effectively balances exploration and exploitation by focusing on grouped hyperparameters, leading to faster convergence to high-performing configurations. This visual comparison supports the conclusion that GSOS offers enhanced search efficiency, reduced fluctuation in accuracy, and quicker identification of optimal hyperparameters, demonstrating the advantages over the Simultaneous approach on some datasets.

## 7. Conclusions and Future Work

This paper presented the Grouped Sequential Optimization Strategy (GSOS) as an enhancement to conventional hyperparameter optimization methods. Leveraging insights from Hyperparameter Importance Assessment (HIA), GSOS sequentially optimizes grouped hyperparameters based on their relative impact on model performance, aiming to improve efficiency in high-dimensional hyperparameter spaces. Our experiments on six diverse image classification datasets demonstrated that GSOS can significantly reduce both the time to find optimal hyperparameters and the total optimization time without compromising model performance substantially. On average, GSOS achieved a 19.69% reduction in time to reach optimal hyperparameters and a 31.9% reduction in total optimization time. Additionally, GSOS displayed faster convergence and greater stability across iterations on some datasets. Despite these advantages, GSOS presents a minor trade-off in accuracy, with slight decreases observed in validation and test accuracies across some datasets. This trade-off may be acceptable in scenarios where the primary goal is to optimize computational efficiency rather than

maximize absolute accuracy. Nonetheless, GSOS offers a promising framework for time-sensitive applications or for cases where model tuning must be balanced against resource constraints.

Future research on GSOS can further enhance its applicability and efficiency by exploring its extension to other model architectures such as Recurrent Neural Networks (RNNs) or Transformer-based models, which would demonstrate the versatility of GSOS across various deep learning applications. Additionally, integrating GSOS with optimization techniques like Hyperband or reinforcement learning could dynamically allocate resources to different hyperparameter groups, enhancing search efficiency. Applying GSOS within AutoML pipelines could also improve automated model tuning, reducing the time to achieve high-performing models in production environments. These avenues suggest GSOS's potential for broad applications and its promise in advancing efficient hyperparameter optimization.

# Acknowledgements

RW gratefully acknowledges financial support from the China Scholarship Council. And MG thanks EPSRC for the support on grant EP/X001091/1.

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

## A. Appendix: Time Analysis of TPE-based Bayesian Optimization

To investigate the computational overhead in TPE-based Bayesian optimization, we decomposed the total optimization time into two main components:

**Model Evaluation Time** ($T_{\mathbf{eval}}$): The time required to evaluate the objective function, which typically involves training and validating a machine learning model.

**TPE Process Time** ($T_{\mathbf{tpe}}$): The time spent by the TPE algorithm to suggest the next sampling point.

The total time per iteration can be expressed as:

$$T_{\text{iter}} = T_{\text{eval}} + T_{\text{tpe}} \tag{8}$$

We simulated a simple model evaluation process and varied the number of hyperparameters in the search space from 1 to 12. The optimization objective is defined as:

$$L(\mathbf{x}) = \text{random loss value}, \quad \mathbf{x} \in \mathbb{R}^d \tag{9}$$

where $d$ is the number of hyperparameters. The objective function returns a random value to simulate loss, and the model evaluation time is approximated by adding a constant delay of $0.01$ seconds per evaluation.

The search space for $d$ hyperparameters is defined as:

$$\mathbf{x} = \{x_1, x_2, \ldots, x_d\}, \quad x_i \sim U(0, 10^6) \tag{10}$$

For each configuration, we performed 100 iterations using the Tree-structured Parzen Estimator (TPE) as the optimization algorithm:

$$\mathbf{x}^* = \arg \min_{\mathbf{x} \in \mathcal{X}} L(\mathbf{x}) \tag{11}$$

The measured TPE process time ($T_{\text{tpe}}$) for different numbers of hyperparameters is summarized in Table 5.

From the results, we observe that the computational complexity of TPE's internal sampling remains efficient even in higher-dimensional search spaces. In our simulated experiments, $T_{\text{eval}}$ was fixed to $0.01$ seconds per iteration. For large-scale machine learning tasks, $T_{\text{eval}}$ can easily range from several seconds to hours, making $T_{\text{tpe}}$ negligible in comparison.

This analysis demonstrates that TPE-based Bayesian optimization is computationally efficient in determining the next sampling point, as $T_{\text{tpe}}$ contributes minimally to the total time. However, the overall optimization efficiency heavily relies on reducing $T_{\text{eval}}$, for instance, by parallelizing model evaluations or employing surrogate models.

Table 5: **TPE Process Time Excluding Model Evaluation**

| Number of Hyperparameters ($d$) | $T_{\text{tpe}}$ (seconds) |
|---|---|
| 1 | 1.267 |
| 2 | 1.496 |
| 3 | 1.569 |
| 4 | 1.618 |
| 5 | 1.662 |
| 6 | 1.818 |
| 7 | 1.762 |
| 8 | 1.884 |
| 9 | 1.820 |
| 10 | 1.815 |
| 11 | 1.590 |
| 12 | 2.050 |

