# OpenReview forum: "Grouped Sequential Optimization Strategy - the Application of Hyperparameter Importance Assessment in Deep Learning"
_CPAL.cc/2025/Proceedings_Track — CPAL 2025 (Proceedings Track) Poster_

### Official Review · Reviewer_VEHC · 2025-01-07
**review for submission 35**

**Rating:** 7
**Confidence:** 3

**Review:**

This paper explores an innovative method of hyperparameter optimization (HPO) that leverages insights from hyperparameter importance assessments (HIA). By grouping hyperparameters based on their impact on model performance, the authors propose a sequential optimization strategy that aims to streamline the HPO process, reducing computational effort and time without compromising model efficacy.

[Strengths]:
1. Innovative Optimization Strategy: The proposed Grouped Sequential Optimization Strategy (GSOS) is a novel approach that systematically prioritizes and optimizes hyperparameters based on their assessed importance, potentially offering significant efficiency improvements over traditional methods.
2. Empirical Validation: The method is validated across multiple datasets, providing a robust statistical foundation that demonstrates the effectiveness of GSOS in reducing optimization time by an average of 31.9% compared to conventional methods.
3. Relevance and Practicality: This research addresses a critical bottleneck in machine learning workflows, making it highly relevant for applications requiring rapid model development and deployment.

[Drawbacks]:
1. Potential Overfitting to Datasets: While the paper presents comprehensive results, the optimization strategy might be overly tailored to the types of datasets used in the experiments, potentially limiting generalizability.
2. Trade-off Between Speed and Accuracy: The paper notes a minor trade-off in accuracy for speed. This could be a concern for applications where even a small decrease in accuracy is critical.

---

### Official Review · Reviewer_Ry9q · 2025-01-12

**Rating:** 6
**Confidence:** 3

**Review:**

This paper introduces a Grouped Sequential Optimization Strategy that leverages Hyperparameter Importance Assessment to enhance HPO efficiency for CNNs. The authors propose grouping hyperparameters based on their importance and sequentially optimizing them using Tree-structured Parzen Estimator-based Bayesian optimization. The experiments, conducted on six image classification datasets, demonstrate that GSOS can reduce the total optimization time by 31.9% on average compared to traditional parallel HPO methods, with only minor reductions in accuracy. The contributions include proposing a novel GSOS framework that sequences hyperparameters for improved efficiency, integrating of GSOS with TPE-based Bayesian optimization, providing empirical validation across diverse datasets, showing significant time savings while maintaining competitive accuracy.

Strengths:
1. The paper introduces a well-motivated approach that extends the utility of HIA by applying it to group and optimize hyperparameters sequentially.
2. Extensive experimentation across six datasets demonstrates the robustness of the approach in different contexts.
3. The reported average time savings of 31.9% without significant accuracy loss is a valuable contribution, particularly for large-scale, time-sensitive applications.

Weaknesses:
1. The reported decrease in validation and test accuracy (e.g., up to 2.23% for validation) may be unacceptable in high-stakes applications where performance is critical.
2. The experiments focus solely on CNNs; extending the analysis to other architectures like transformers could strengthen the generalizability of the proposed method.
3. The approach assumes a static ordering of hyperparameters, which may not scale well for more complex or dynamically changing search spaces.

---

### Official Review · Reviewer_fdzf · 2025-01-15
**a novel hyperparameter optimization approach called Grouped Sequential Optimization Strategy (GSOS).**

**Rating:** 6
**Confidence:** 2

**Review:**

This paper explores a new hyperparameter optimization (HPO) strategy named “Grouped Sequential Optimization Strategy” (GSOS). Instead of tuning all hyperparameters simultaneously (as is common in standard Bayesian optimization), GSOS clusters hyperparameters according to their measured importance and then optimizes each group in sequence.

Strengths:

(1) Clear Motivation and Relevance: the paper addresses a well-known bottleneck in deep learning: hyperparameter tuning can be computationally expensive, especially for high-dimensional search spaces.

(2) the proposed GSOS strategy is novel.

Weakness:

(1) Limited evaluations. Evaluations on other deep architectures (like Transformers, large language models, or RNN-based models) would help demonstrate broader applicability.

(2) Limited Discussion on Accuracy Trade-Off. A deeper discussion of when such a trade-off becomes unacceptable or how it could be mitigated would strengthen the paper’s insights.

---

### Meta-Review · Area_Chair_FVhb · 2025-02-06

**Recommendation:** Accept (Poster)
**Confidence:** 4

**Metareview:**

This paper proposes a novel application of hyperparameter importance assessment, i.e. grouping hyperparameters sequentially according to their importance and then optimizing over every group rather than individual parameters. All the reviewers appreciate the idea and its potential, while noting that the empirical evaluation was a bit limited to specific datasets and architectures.

---

### Decision · Program_Chairs · 2025-02-11

Accept (Poster)